# Classifying Nuclei Shape Heterogeneity in Breast Tumors with Skeletons

**Abstract.** In this study, we demonstrate the efficacy of scoring statistics derived from a medial axis transform, for differentiating tumor and non-tumor nuclei, in malignant breast tumor histopathology images. Characterizing nuclei shape is a crucial part of diagnosing breast tumors for human doctors, and these scoring metrics may be integrated into machine perception algorithms which aggregate nuclei information across a region to label whole breast lesions. In particular, we present a low-dimensional representation capturing characteristics of a skeleton extracted from nuclei. We show that this representation outperforms both prior morphological features, as well as CNN features, for classification of tumors. Nuclei and region scoring algorithms such as the one presented here can aid pathologists in the diagnosis of breast tumors.

**Keywords:** Medial axis transform, breast cancer, digital pathology, computer vision

## 1 Introduction

Recent advancements in whole-slide imaging technology have paved the way for what has been termed Digital Pathology, the digitization of pathology data. Once a biopsy sample has been processed, it can then be scanned into a digital format for viewing by the pathologist. This allows for viewing at any time, while not affecting diagnostic accuracy [1]. Additional benefits of Digital Pathology include the ability to store large amounts of cases for teaching and research purposes, as well as facilitating telepathology, allowing multiple pathologists to view cases simultaneously and remotely. However, what we are most interested in is how the digitization of pathology data allows for the use of computational algorithms to analyze the data and provide useful information to the pathologist.

Digitization proceeds as follows. Slices of the extracted tissue sample are scanned into whole slide images (WSI's). These images are far too large and high resolution (around 5 gigapixels) to perform most analytical computation on, so regions of interest (ROI's) may be extracted via a plethora of methods [2]. Regions are often extracted via texture analysis, where regions of high texture variance are more likely to contain useful information, like a breast duct. These smaller sub-regions of the whole slide (usually around 1M pixels) are likely to contain information useful to the diagnosis, while ignoring large parts of the image which are not useful (such as white space). Most computational analysis is

done on the region level. With these manageable ROI's, we can employ a variety of machine learning and computer vision techniques to guide pathologists in the decision making process.

This study focuses on breast tumor analysis in particular for several reasons. Breast cancer is the second most common form of cancer found in women in the US, with an estimated 12% of women developing invasive breast cancer over the course of their life [3]. Annually, roughly 1.6 million breast biopsies are conducted, with around 25% of them showing malignancy [4]. The remaining cases may be classified along a spectrum of labels, from fully benign up to a very high risk of becoming malignant in the future. The differences in appearance between these classes of pre-malignant tumors can be incredibly subtle, which is reflected in the high rates of discordant diagnoses among pathologists for those high-risk cases [4]. In particular, discordance rates are 52% for high-risk benign lesions, compared to just 4% for distinguishing between invasive and non-invasive cancer [4]. Despite the challenges, accurate diagnosis of these lesions is crucial in providing optimal treatment, and early detection and treatment reduces the risk of future malignancy [5–7].

Differentiating between a benign lesion and a low or high-risk lesion requires analysis of subtle structural and shape changes in the breast tissue. Complex, high-level structural changes occur in the tumor region as it progresses along the spectrum from benign to malignant [8]. High-risk lesions often show breast ducts being crowded with nuclei in rigid patterns around the lumen (interior opening of the duct). Even the smallest structures present in the tissue, the cell nuclei, demonstrate a *change in shape* as the disease progresses along the spectrum from benign to malignant.

In particular, *higher-risk* lesions, like flat epithelial atypia (FEA) have *rounder* epithelial nuclei than lower risk lesions like columnar cell change (CCC) [9]. Atypical ductal hyperplasia (ADH) is one of the highest risk tumors. Epithelial nuclei in ADH tumors are typically completely rounded [10]. Further, to get a full picture of the tumor environment in order to make an accurate prediction, one must consider not only the distribution of structures across the image (how densely the nuclei are crowded and in what pattern), but also the shape of the structures themselves, in particular the nuclei. This is the focus of our study, to find an accurate and robust way of characterizing the shape of epithelial nuclei in breast tumors.

In this study, we apply the Medial Axis Transform (MAT) [11], a skeletonization algorithm, to the task of classifying nuclei as tumor or non-tumor based solely on their shape. We propose a variety of features to aggregate MAT per-pixel scores into region scores, including novel features measuring branching, bend, and eccentricity. Using our features extracted from the skeleton, we show considerable separation of the two classes of nuclei. In particular, we show that our MAT-based features allow more accurate cancer classification, compared to prior morphological features and those extracted from a CNN applied on a color image or one applied on a 3-channel skeleton image.

We highlight the importance of model interpretability in medical studies such as this one. Many deep learning models exist which obtain impressive classification results in this domain, such as in [12] where they achieve almost 98% accuracy on the BreakHis dataset (benign vs. malignant tumors). However, one thing missing from deep learning models is the ability to directly interpret results. Methods exist for probing deep nets to obtain some result interpretability, such as LIME [13], but these require further processing. This is especially important when considering who will be using the software, primarily pathologists, who are not experts in data science. If our aim is to improve the field of pathology by providing pathologists with tools that will aid them in their diagnostic process, a crucial step is gaining the trust of pathologists, and that can be accomplished through model interpretability. Using hand-picked features which can be mapped to real world descriptions (such as a nucleus being rounded) and simpler decision models leads to better potential interpretability and more trust in the model. Our method thus explores a shape representation that achieves competitive results but is also interpretable. Further work could include adding interpretability into the proposed feature extraction framework.

## 2   Related Works

Although classification of *whole* breast tumors is a more common problem, several studies have been conducted on classifying *individual* nuclei within the tumor images. Although the end goal is to be able to classify whole tumor images, studying the individual nuclei in the image can be a useful sub-task, as tumor nuclei change shape as a tumor progresses. Being able to highlight individual potentially problematic nuclei in a tumor image would also lend to a more interpretable model. Models that seek to classify from the whole image without special attention to the nuclei, as well as other biological structures, are ignoring biological precedent for the classification. The classic Wisconsin Breast Cancer dataset [14], which features nuclei labeled as either malignant or benign, along with 10 hand-computed features for each nucleus (e.g. radius, perimeter, texture), was used to train simple classifiers with near 100% accuracy on a small test set, such as in [15]. However, a model trained on these parameters could never be used in a clinical setting, as it would require the manual measurement of each nuclei in the biopsy scans, a task more expensive and time consuming than having a pathologist look at it manually.

Many methods exist which seek to replicate the above results in a more scalable fashion, with the nuclei features being computed by an algorithm. Once these explicitly defined features are extracted, simple classifiers can be trained on them with reasonable success. In [16], the authors extracted four simple shape features and 138 textural features. They then classified the nuclei into benign and malignant using an SVM, achieving 96% accuracy. Similarly, in [17], the authors extracted 32 shape and texture features and classified the nuclei into high and low risk categories using an SVM with an accuracy of 90%. One downside of taking this approach to nuclei classification is the requirement of manually obtaining

or computing nuclei segmentations, or boundaries around each nuclei, in order to do the shape and texture analysis.

Methods which exploit deep learning do not strictly require a segmentation. For example, Raza [18] use a CNN to classify colon tumor nuclei into four classes with state-of-the-art results. Many works exist which study various CNN frameworks on classifying whole tumor images, such as in [19–21], all of which show very promising results. However, the main issue with deep learning approaches is a lack of inherent interpretability. Pathologists may be reluctant to incorporate such black-box methods into their workflow. Although methods exist which allow us to interpret results regardless of the model (such as LIME [13]), there is motivation to keep models more simple and inherently interpretable.

## 3    Approach

We first describe the data we use, and how we pre-process it. Next, we describe our approach that leverages the well-known Medial Axis Transform (MAT) [11]. Finally, we relate our full set of novel features that we extract from the MAT skeleton.

### 3.1    Data

We use the *BreCaHAD* dataset of [22], which contains labeled tumor and non-tumor nuclei (i.e. malignant nuclei present inside a malignant tumor and nuclei outside of the tumor, in the stroma, etc.). We leave studying the finer-grained nuances in the middle of the spectrum (i.e. classifying high-risk benign lesions) as future work, because no such annotated dataset is currently available. *BreCaHAD* consists of 162 breast lesion images, all of which show varying degrees of malignancy. The centroid of each nuclei in the images is labeled with one of the following six classes: mitosis, apoptosis, tumor nuclei, non-tumor nuclei, tubule, and non-tubule. For our purposes, we are just interested in differentiating the tumor and non-tumor nuclei. Theoretically, if we can design a framework to differentiate tumor and non-tumor nuclei in malignant cases based on shape, the same framework should be successful at analyzing high-risk benign tumor nuclei, the more challenging task (for which no annotated datasets currently exist).

The dataset presents some challenges. Most importantly, all of the images show malignant tissue samples. Thus, at the nuclei (rather than full tissue sample) level, there is a dramatic class imbalance in the data, with almost 92% of nuclei labeled as 'tumor'. Additionally, the ground truth labels (centroid and class label) are not enough for our purposes since nuclei need to be segmented first for further processing (feature extraction and classification). We next describe how we handle both challenges.

*Nuclei segmentation:* If we wish to analyze the shape of the nuclei present in the biopsy images, we need some definition of the boundaries of the nuclei. Ideally, this would come with the dataset, and be a ground truth value. However, in our case, we are only given centroid locations of the nuclei, and thus must

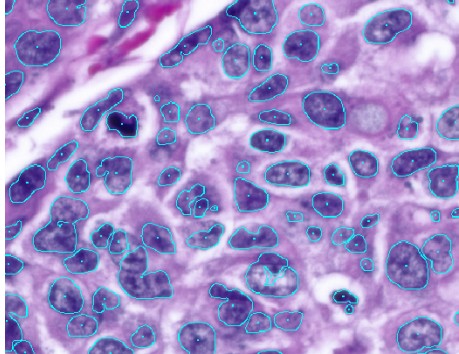

**Fig. 1.** Segmentation results using the method described in [23]. Overall the output is fairly accurate, but some of the nuclei are segmented together, into a larger blob.

produce our own segmentation. We use the segmentation algorithm proposed by Chen [23], which is a fully convolutional network based on U-Net [24], with additional smoothing applied after the upsampling. A segmentation on a sample from the dataset using this method is shown in Figure 1. Once the segmentation is computed, the labeled centroids can be matched to their corresponding nuclei boundary by determining which (masked) boundary the centroid falls within. The labels in the dataset could then be associated with a nuclei boundary. Using this segmentation scheme, 80.18% of the ground truth nuclei in the dataset were found (18149 nuclei). Of these, 92% were positive (16689). Crops of size $(81, 81)$ pixels centered at each nuclei were taken from the original images, and used as the training and testing data.

*Class imbalance:* To account for the class imbalance, training and testing sets were obtained by randomly sampling with replacement from the over-represented class (tumor) until there was an equal number of samples for both classes. Without this step, the model would tend to demonstrate very low specificity (high rate of false positives), while still maintaining a low loss. There were 1460 negative (non-tumor) nuclei after segmentation, so the same number of positive nuclei were randomly selected from the initial set of 16689. This is not ideal, as many positives are thrown out, however it is preferable to an overfitted model.

### 3.2    Medial Axis Transform

We propose a novel representation for nuclei shape analysis, based on the medial axis transform (MAT) [11]. The MAT is a skeletonization algorithm wherein a closed binary shape is reduced down to a 1-pixel-wide branching structure wholly contained within the original shape. This may be conceptualized as a thinning process, where the border of the shape becomes thinner, or erodes, until opposite sides of the shape come together, defining the skeleton. The skeleton points are center points of maximally inscribed discs, i.e. circles with more than one point on their surface tangent to the shape boundary. This property makes the MAT

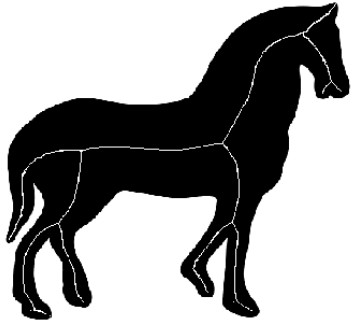 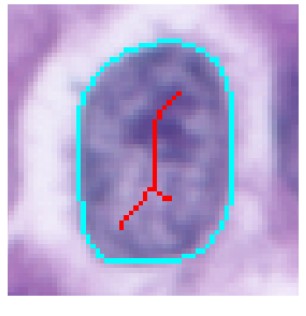

**Fig. 2.** Medial Axis Transform on a binary image of a horse (left) [18]. A nuclei segmentation (cyan) and skeletonization (red) on a tumor nucleus is shown on the right.

skeletonization an appropriate candidate for deriving a shape descriptor. We show an example in Figure 2.

The skeleton itself does not contain enough information on its own to derive a shape descriptor, as it only provides us with a set of pixels which define the skeleton. However, it can be used to derive several scoring statistics. Rezanejad [25] derive three scores from the MAT: ribbon, taper, and separation. They also show that these scores can be successfully used by a CNN to perform scene classification from line images (no color or texture). The scene classification task is significantly different from the nuclei classification one, as scene images feature many contours, some of which may not be closed shapes. Further, a nuclei image will contain far fewer contours compared to a dense scene. However, the results in [25] imply that these scores capture contour shape accurately, which applies to nuclei shape analysis, where we are hoping to capture how rounded the nuclei are.

Let $p$ be a point on the skeleton defined by its pixel location in the image, $R(p)$ be the radius function for that point (shortest distance from the skeleton point to the boundary), and $[\alpha, \beta]$ be a range of skeleton points of size $k$. Finally, let $p \in [\alpha, \beta]$. Then the three score metrics are defined as follows, based on the first and second derivative of $R(p)$:

$$S_{ribbon}(p) = R'(p) \tag{3.1}$$

$$S_{taper}(p) = R''(p) \tag{3.2}$$

$$S_{separation}(p) = 1 - (\int_\alpha^\beta \frac{1}{R(p)} dp)/k \tag{3.3}$$

Because pixels are discrete, and thus so are the intervals we are integrating and deriving over, we must use numerical gradients. A small value of $k$ was used, as the nuclei are fairly small (roughly 100 pixel perimeter), and the scoring metrics are designed to capture local symmetry. For all of the tests, we used a value of k=8. The skeletonization method was adapted from [26–29].

**Table 1.** Proposed skeleton features

| Feature | Description |
|---------|-------------|
| Min rib/tap/sep | Minimum of each of the 3 scores for all pixels in the skeleton (ribbon, taper, separation) |
| Max rib/tap/sep | Maximum of each of the 3 scores |
| Mean rib/tap/sep | Mean of each of the 3 scores |
| Deviation rib/tap/sep | Standard deviation of each of the 3 scores |
| Max branch length | The length (in pixels) of the longest branch of the skeleton |
| Avg branch length | Average length (in pixels) of the branches of the skeleton |
| Num branches | The number of branches in the skeleton |
| Bend | Angle between the line connecting the furthest two points on the skeleton and the skeleton's major axis |
| Major/minor axis len | Length (in pixels) of the major and minor axes of the ellipse with the same normalized second central moments as the skeleton |
| Eccentricity | Eccentricity of the ellipse that has the same second-moment as the skeleton, the ratio of the distance between the foci of the ellipse and its major axis length |
| Solidity | Proportion of pixels in convex hull that are also in the skeleton |

The Rezanejad algorithm defines three scores for each skeleton point. Thus, it could be treated as a 3-channel color image, where all non-skeletal pixels have a value of 0 and skeleton points are described with 3 scores. The ribbon score ($S_{ribbon}(p)$) captures the degree of parallelity of the surrounding contours, and increases as they become more parallel. The taper score ($S_{taper}(p)$) is designed to increase as contours resemble the shape of a funnel or railroad tracks, but also has a high value for parallel contours. Separation ($S_{separation}(p)$) captures the degree of separation between the contours, and increases with distance.

### 3.3   Final Representation

We propose a set of 20 features, described in Table 1. Three of those features rely on the per-pixel ribbon, taper and separation features described above. The per-pixel features are aggregated on the whole nuclei level by taking their min, max, mean, and standard deviation.

The following additional skeleton structural features were extracted: number of branches, average and max branch length, and bend (angle between line connecting furthest 2 skeleton points and major axis). Branches were isolated by computing junction points using [30], then setting junction points to zero and applying connected component analysis on the separated branches.

Another interesting method for extracting a feature from the skeleton data would be to define the skeleton as a graph, with node features being the 3 scores, and apply graph convolutions or graph kernels to do the classification. Deep learning on graph structured data has shown promise in recent years [31]

and could prove useful in this task. However, the graph analysis is outside of the scope of this study currently.

## 4    Experimental Validation

We describe the methods compared and metrics used. We next show quantitative results, and finally some examples of the features that our method relies on.

### 4.1    Methods compared

In order to show the efficacy of the proposed method for capturing nuclei shape, the tests will be conducted on features with no information of texture or color in the nuclei. For this reason, we test our method against another method which does not incorporate texture, but has shown to be successful. Yamamoto [17] obtained an accuracy of 85% using only a small number of hand-picked morphological features and an SVM. We were able to replicate their accuracy results on the balanced test set using the following set of features originally proposed in [17]: area, eccentricity, extend, major axis, minor axis, convex area, circularity, equivalent diameter, filled area, perimeter, and solidity. Note again how none of these features incorporate any color or texture information. In total, 19 morphological features were extracted, similar to the dimensionality of the skeleton feature vector we will be comparing it to. We refer to this method as **Morphological features** in Table 2.

However, color and texture features have also been used with success [16]. Thus, we also compare the proposed shape-based method to a standard convolutional neural network which includes color, namely AlexNet [32], pre-trained on ImageNet. We used this model to extract features from the last fully-connected layer (D=1000). The CNN was tested both on the full color crops, as well as the 3-channel skeleton score image (see below). We refer to these methods as **CNN on full color crop** and **CNN on 3-channel skeleton image** in Table 2, respectively.

Our method uses the 20-dimensional proposed feature vector described in Section 3.3, and is referred to as **MAT skeleton features**.

### 4.2    Metrics and setup

We evaluate model performance using accuracy, sensitivity, and specificity, with special attention paid to sensitivity. In medical tests, maximizing sensitivity, or minimizing false negatives, is an important goal, as we do not want to under interpret a sample and cause a patient to not receive necessary treatment. The dataset was split into 70% for training, and 30% for testing, giving sizes of 2044 and 876 nuclei respectively. Classification for all tests was done using a linear SVM with RBF kernel. We use an equal class representation, so a random algorithm would achieve an accuracy of roughly 0.5. Scores were averaged over 20 runs, with a different random train/test split for each run.

**Table 2.** BreCaHAD Classification: Main Results

| Method | Accuracy | Sensitivity | Specificity |
|---|---|---|---|
| Morphological features | 0.846 +/- 0.001 | 0.831 +/- 0.001 | **0.861 +/- 0.001** |
| CNN on full color crop | 0.632 +/- 0.001 | 0.477 +/- 0.001 | 0.798 +/- 0.001 |
| CNN on 3-channel skeleton image | 0.654 +/- 0.002 | 0.623 +/- 0.001 | 0.687 +/- 0.001 |
| **MAT skeleton features (Ours)** | **0.850 +/- 0.004** | **0.844 +/- 0.004** | 0.852 +/- 0.003 |

### 4.3   Quantitative results

Our results are shown in Table 2. Our proposed method achieves the highest accuracy. Further, the results are statistically significant. We reject the null hypothesis that the distributions of accuracies for the skeleton features and the morphological features come from different distributions at significance level 5%.

Additionally, our MAT skeleton features achieve the highest sensitivity of the methods testing. Sensitivity is especially important in a medical setting, where a false negative (missed disease diagnosis) is much more costly than a false positive. This also highlights the failure of the color CNN feature, where its passable accuracy score was achieved by outputting mostly negative labels, resulting in many false negatives. It also demonstrates that the 3-channel skeleton image CNN feature was actually significantly better than the CNN on the full color image, as is seen in the difference in sensitivity. Note that we could have tried a larger CNN for extracting the features, but the sizable gap in the CNN methods' accuracy vs. ours and the morphological features, indicated that further exploration of CNN features may not work well.

Finally, we compare performance of the SVM on the MAT skeleton scores feature which closely follows [25] (min/max/mean/deviation of the ribbon, taper, and separation scores) and the new skeleton features that we propose (max branch length, number of branches, etc.). The results can be seen in Table 3. While the MAT score-derived features are more successful in the classification of the nuclei shape in isolation, the other features we propose do significantly contribute to performance.

Scores could likely be improved for both the morphological and skeleton features by taking several steps. First, color features could be incorporated into the skeleton and morphological features. These could be extracted from a CNN, or just be basic statistical measures such as in [16]. However, with these results we have shown the efficacy of the MAT skeleton feature as a means of distinguishing nuclei shapes.

### 4.4   Qualitative results

Analysis of the linear predictor coefficients of the SVM with all MAT skeleton features in Table 4 shows that the maximum separation score feature was by far the most discriminant. Recall that the separation score measures the level of

**Table 3.** BreCaHAD Classification: Ablation Results

| Method | Accuracy | Sensitivity | Specificity |
|---|---|---|---|
| Min/max/avg/dev skeleton features | 0.820 +/- 0.003 | 0.789 +/- 0.011 | **0.853 +/- 0.006** |
| Other skeleton features | 0.692 +/- 0.002 | 0.680 +/- 0.004 | 0.705 +/- 0.009 |
| **All skeleton features (Ours)** | **0.850 +/- 0.004** | **0.844 +/- 0.004** | 0.852 +/- 0.003 |

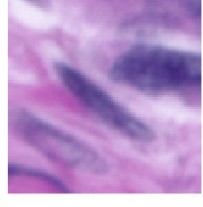 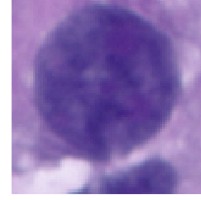

**Fig. 3.** The nuclei with the lowest maximum separation score (left), a non-tumor nuclei, and that with the highest maximum separation score (right), a tumor nuclei.

separation between the skeleton and its boundary. Intuitively, this makes sense, as we know tumor nuclei to be more engorged and rounded (and thus have high separation), and non-tumor nuclei to be thinner and more elongated (resulting in a smaller separation). This intuition is validated by viewing the data samples which exhibit the highest and lowest max separation scores for the respective classes. The sample which exhibited the highest max separation score was from the positive class, and the sample which exhibited the lowest separation was negative, both of which are illustrated in Figure 3. Other important features include our new axis and branch features.

## 5    Conclusion

The primary aim of the study was not to achieve the best classification metrics and outperform all other state of the art models, but rather to define a robust shape descriptor that accurately captures the change of shape of the nucleus. The MAT skeleton feature showed improvement over the baseline morphological features. Additionally, features extracted with a CNN on the weighted skeleton image performed better than similar features extracted from the full color image. The MAT skeleton scoring algorithm outlined in this paper is thus a useful shape descriptor in regards to tracking nuclei shape heterogeneity in breast tumors.

Although here the MAT skeleton feature was applied to distinguishing tumor and non-tumor nuclei in malignant breast tumors, it may be robust enough to tackle harder challenges, such as the classification of high-risk tumors, or be applied to other modalities and cancers, such as prostate cancer (which also uses stained slide images for diagnosis). Further work will focus on applying this technique to more challenging tasks and datasets.

**Table 4.** SVM Beta Values for the top 5 features

| Feature | Beta Value |
|---|---|
| Max separation | 4.082 |
| Major axis length | 0.844 |
| Average branch length | 0.582 |
| Number of branches | 0.580 |
| Minor axis length | 0.519 |

We can also apply the nucleus scoring algorithm, where each nuclei receives a set of features, to classify a whole region by considering all contained nuclei. One approach is to take a majority vote: if most nuclei in the region are cancerous, then predict the image as cancerous. Another method could be normalized mean statistics being taken over the graph to return a 1-dimensional vector. A more interesting way to combine the nuclei features across the image could be to define a graph over the image, where the nodes in the graph are the nuclei, and each node inherits the features describing the nuclei. This would maintain the structure and distribution of nuclei in the original image, while allowing for various graph learning methods to be applied, which can consider both the individual nuclei features, as well as the overall structure of the tumor environment. This is a topic for further study.

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
