# OpenReview forum: "Classifying Nuclei Shape Heterogeneity in Breast Tumors with Skeletons"
_thecvf.com/ECCV/2020/Workshop/BIC — BIC 2020 Oral_

### Official Review · AnonReviewer2 · 2020-07-29
**A promising method, but the reliance on the initial segmentation is not sufficiently explored**

**Rating:** 6
**Confidence:** 3

**Review:**

This work introduces the use of the properties of skeletonised nuclei for classifying whether nuclei should be classified as tumour or non-tumour in the context of breast cancer diagnosis.

### Quality
The work should be commended for taking care to discuss the aims, limitations, and potential extensions of the approach. The authors also acknowledge the limitations of the test data set and steps taken to mitigate bias. While the MAT skeleton features method introduced by the authors does not numerically outperform other methods in measures of accuracy or specificity, it offers a noticeable improvement in sensitivity. The major criticism that I have with regard to quality is that the use of skeletonization as a descriptor is heavily dependent on the initial segmentation of the nuclei in the image. Although the authors have performed ablation to disentangle the contribution of different skeleton descriptors to overall accuracy/sensitivity/specificity, the impact of initial segmentation quality was not taken into consideration. This is slightly worrisome as this is the foundation upon which the entire method is based, and the authors themselves in the caption of Figure 1 describe the segmentation performance as only ‘fairly accurate’. Again, the caption of Figure 1 acknowledges the merging of nuclei together, which will then of course dramatically alter the skeleton(s) in that region. An interesting experiment would have been, for example, to compare results between an expert manual segmentation for one or two regions of interest with the U-net based segmentation.

### Clarity
On the whole the paper is very clearly written and presented and is easy to read and comprehend. However, there are a few instances where this is not the case. There is a little confusion with regard to the advantages of this approach compared to related works. Specifically, the authors explain that segmentation of nuclei is a downside (lines 133-136), yet their approach necessarily requires segmentation. A major shortcoming in clarity is in how the method deals with nuclei that are annotated as neither tumour nor non-tumour (i.e. the four additional classes listed in lines 163-164); this is not discussed. A diagrammatic representation of the features in Table 1 may have added value to the paper, as would have addition of the segmentation boundaries and skeletons to the images in Figure 3.

### Originality
The authors do not introduce any novel algorithms or metrics, rather they integrate established segmentation methods (modified U-net), transforms (medial axis transform), and descriptors (ribbon, taper, separation). The application to classification of breast biopsy nuclei does appear to be novel.

### Significance
One of the biggest strengths of this paper is the focus on what is necessary in an image analysis tool for pathologists diagnosing breast cancer. The authors clearly outline the need for interpretable and tangible metrics (e.g. shape descriptors rather than the ‘black box’ of deep learning methods), and they clearly weigh the benefits of this interpretability against methods that may ‘score’ better. This strength (in my opinion) outweighs the need for a leap in computational novelty.

### Pros
* The work is focused towards a specific application and integrates the needs of that application (both from an analytical standpoint and a practical standpoint) into the method
* The performance of the method is generally good
* The added value, limitations, and future extensions of the skeleton-based approach are explored and discussed.

### Cons
* The method is necessarily dependent on the initial segmentation quality, and this is not sufficiently explored
* The method performs a binary classification on the detected nuclei (tumour/non-tumour) when in fact there are six possible classes in the ground truth data; it is unclear how this is resolved in this method.


**Reviews Visibility:**

I agree that my anonymized review is made publicly visible, if the submission is accepted.

---

### Official Review · AnonReviewer1 · 2020-07-31
**Classifying Nuclei Shape Heterogeneity in Breast Tumors with Skeletons**

**Rating:** 6
**Confidence:** 5

**Review:**

Summary:
This paper addresses the problem of nucleus classification based on its morphological characteristics. The problem is motivated by the need to automate analyses of ROIS extracted from whole slide images and classifying the ROIs into tumor and non-tumor regions based on the shapes of nuclei. The authors approach the problem by
•	detecting boundaries of nuclei in ROIs,
•	verifying the boundaries against ground truth centroids in BreCaHAD dataset from the paper accessible at  https://bmcresnotes.biomedcentral.com/track/pdf/10.1186/s13104-019-4121-7,
•	applying a medial axis transform (MAT) to nuclei,
•	defining 20 features derived from MAT-based skeletons, and
•	classifying the ROIs based on the features using SVM with RBF kernel.
The experimental results include comparisons of the proposed method against other subsets of features and CNN-based methods applied to raw images.

Strengths:
The method for preparing training data based on the available BreCaHAD dataset is very creative
The 20 features derived from MAT-based skeletons is novel.

Weaknesses:
The authors argue at the beginning about the importance of model interpretability. However, that argument is never mentioned in the experimental section. The reader would expect the argument to be developed in Table 2 (add a column for model interpretability). In addition, the authors should demonstrate that they designed all features to have a physical meaning while many other man-constructed features can have very hard to interpret meaning, for example, many texture features (GLCM, wavelet, Zernike, etc.) are very hard to explain to biologists.
The paper would benefit from a discussion about feature construction (i.e., physical meaning, link between the biological observations and feature construction), feature dimensionality (i.e., why 20 features?), and feature selection (i.e., ranking of all features from all methods as shown in Table 4 for top 5).




Main Comments:
The paper would really benefit if you could create synthetic data demonstrating the performance of individual features for a range of nucleus shapes
The work would also be better received if you would create a visualization (as hinted in your conclusion section) that has a pseudo-colored spatial graph of nuclei. The reason for my comment is that your classification can take into proximity information depending on whether tumor nuclei are interleaved with non-tumor nuclei or tumor nuclei form spatially isolated colonies.



**Reviews Visibility:**

I agree that my anonymized review is made publicly visible, if the submission is accepted.

---

### Decision · Program_Chairs · 2020-07-31

Accept (Oral)